# Hypothesis: Cancer Hormesis and Its Potential for Cancer Therapeutics

**DOI:** 10.3390/life14030401

**Published:** 2024-03-18

**Authors:** Michael Bordonaro, Darina Lazarova

**Affiliations:** Department of Medical Education, Geisinger Commonwealth School of Medicine, Scranton, PA 18509, USA; dlaz66@aol.com

**Keywords:** cancer, concomitant resistance, hormesis, metastasis

## Abstract

Primary tumors can inhibit the growth of secondary lesions, particularly metastases, in a phenomenon termed “concomitant resistance”. Several mechanisms have been proposed for this effect, each supported by experimental data. In this paper, we hypothesize that concomitant resistance is a form of hormesis, a biphasic dose response in which a stimulus has a positive and/or stimulatory effect at low dosages and a negative, inhibitory, and/or toxic effect at higher dosages. When this paradigm applies to tumorigenesis, it is referred to as “cancer hormesis”. Thus, low numbers of benign neoplastic cells or less tumorigenic malignant cells may result in resistance to the development of malignant neoplasms, including metastases. A host containing a number of (less tumorigenic) neoplastic cells may exhibit greater protection against more tumorigenic malignant neoplasms than a host who lacks neoplastic cells, or who has too few neoplastic cells to stimulate a protective response. As a theoretical endeavor, this paper also proposes that cancer hormesis can be leveraged for therapeutic purposes, by the implantation of safely controlled, benign artificial tumors in high-risk patients. These tumors would prevent the development of endogenous malignant neoplasms by creating an inhibitory environment for such growth. Strategies for testing the hypothesis are proposed.

## 1. Introduction

It has long been known that primary tumors can inhibit the growth of secondary lesions [1,2,3,4], particularly metastases, in a phenomenon termed “concomitant resistance” [3,5,6,7,8,9,10,11]. Concomitant resistance is not universal, and sometimes removal of primary tumors represses the development of secondary neoplasms. However, some of these contradictory findings may be due to the observation that there are two distinct periods in the lifecycle of a primary tumor at which concomitant resistance occurs [7]; the secondary phases of concomitant resistance may be linked to secreted factors that are independent of direct effects on immunomodulation [3,11]. Another possible explanation for the contradictory findings is the presence or absence of established metastases. In the absence of metastases, or the absence of a significant metastatic load, primary tumor removal can be beneficial or even curative; however, established metastases, when present, can demonstrate enhanced progression after excision of the primary neoplasm [3]. Thus, depending on when and how concomitant resistance is assayed, variable outcomes might be observed.

Despite these complications, concomitant resistance is an established and consistently observed phenomenon, and its mechanisms have been researched in considerable detail. For example, there is evidence that the removal of primary tumors can, in many, albeit not all cases, spur additional neoplastic (including metastatic) growth because the inhibitory activity of the primary tumor on secondary lesions is relieved. The clinical consequences of this phenomenon have been evaluated in some detail with respect to colon cancer [3]. Thus, in patients with evidence of established metastases, postoperative enhancement of those metastases was observed, resulting in increased morbidity and mortality. In such cases, immediate chemotherapy coupled with leaving the primary tumor in place is a clinical option assuming that direct morbidity (e.g., obstruction, bleeding) due to the presence of the primary tumor is not a limiting factor. Data for other cancers, while not always as definitive as that of the colon, demonstrate similar outcomes. Thus, surgery for melanoma, as well as for testicular and ovarian cancer, can lead to enhanced distant/metastatic disease [3]. More recent data for other forms of cancer demonstrate enhanced metastasis or increased numbers of circulating tumor cells after removal of the primary tumor, including in cancers of the lung, breast, prostate, and pancreas [4].

Several mechanisms have been proposed to explain concomitant resistance [7]. These include but are not limited to (a) resource competition between neoplastic cells, including competition for nutrients, a situation termed “atrepsis” [3]; (b) secretion, by the primary neoplasm, of anti-proliferative factors, such as tyrosine isomers that influence MAP/ERK and STAT3 signaling and affect cell cycle checkpoints [7]; (c) “concomitant immunity” [11] by which stimulation of the immune system by the primary tumor targets secondary lesions and affects other immune system-related processes, such as those mediated by monocyte-NK cell signaling involving IL15 [10]; (d) changes in inflammation and altered regulation of neuroinflammation [4]; and (e) inhibition of angiogenesis [8], thus directly starving competing neoplasms of oxygen and nutrients. While immunity-related mechanisms have long been favored to explain concomitant resistance, this phenomenon also occurs in immunodeficient animal models; therefore, other mechanisms must also play a significant role [7]. As noted above, the molecular mechanisms of concomitant resistance include secreted factors [11] that affect anti-proliferative signaling, inhibition of angiogenesis, as well as modulation of host immunity.

Tumors possess the ability to control their own growth; this activity is derived from normal growth control mechanisms that are mostly suppressed during tumorigenesis but are still potentially active [8]. If these mechanisms of self-inhibition involve secreted factors [3,11] and these are systemically present, they can also repress the growth of secondary lesions (e.g., metastases). In this manner, tumors can not only leverage aberrant use of normal growth mechanisms to fuel their own expansion but also to inhibit that of neoplastic cells competing for resources in the same host environment. Removal of primary tumors not only affects inflammation but can also increase the numbers of circulating tumor cells [4], suggesting repressive effects on these factors when the primary neoplasm is present.

Inhibition of angiogenesis is a powerful inhibitory mechanism, and well suited for the repressive effect of a primary tumor on distant metastatic growths; however, in silico modeling of the process suggests that inhibition of angiogenesis alone is insufficient for stable inhibition of secondary tumorigenesis [8]. Therefore, concomitant resistance is likely due to all the aforementioned mechanisms, as well as other unidentified mechanisms of action. It is this combination of mechanisms working in tandem that makes concomitant resistance appealing to be leveraged as a preventive or therapeutic approach against human cancer.

## 2. Cancer Hormesis

We posit that concomitant resistance in cancer, and its proposed mechanisms of action, can be explained as a hormesis response. Hormesis is a biphasic dose response in which a stimulus has a positive and/or stimulatory effect at low dosages and a negative, inhibitory, and/or toxic effect at higher dosages [12,13,14,15]. The observation that low dose radiation is possibly beneficial in certain circumstances is a classic example of the hormesis response [13]. Hormesis has been considered as a potential factor in cancer prevention as well as therapeutics [16,17,18] and has therefore been evaluated with respect to treatments for breast cancer [19], radiation therapy [20], combinatorial therapy [21], as well as with the use of dietary compounds for cancer prevention and therapy [22]. However, these studies and proposals deal with either treating an established cancer or using exogenous agents for preventive purposes. In contrast, the question considered in the current manuscript is to determine how neoplastic cells exert concomitant resistance via mechanisms of hormesis. What role does hormesis play with respect to the endogenous development of cancer itself and to its potential metastatic spread?

This paper defines “cancer hormesis” as occurring when the presence of low levels of (relatively benign) neoplastic cells results in a positive anti-malignancy effect for the patient, in part by stimulating immune responses, anti-angiogenic and anti-proliferative signals, as well as other, as of yet unidentified, mechanisms. In contrast, a greater (and/or more malignant) neoplastic load increases patient morbidity and mortality (Figure 1). 

It is therefore expected that an individual with benign neoplastic cells will be more resistant to malignant neoplastic growth arising elsewhere compared to an individual devoid of any neoplastic cells. This latter individual would be a “free and open niche space” for malignant tumors, lacking the concomitant resistance conferred by (less malignant) neoplastic cells that inhibit malignant tumorigenesis.

It is unlikely that any human adult is completely free of neoplastic cells during their lifetime. In the presence of tumor immunosurveillance [23], adult humans likely produce neoplastic cells [8] that are detected and destroyed by the immune system; only when this process fails do tumors develop. At any given time, neoplastic cells may exist before being eliminated by the immune system as these cells are continuously being produced. Further, it is possible that benign tumors that arise from the failure of the immune system to eliminate neoplasia also suppress malignancy via concomitant resistance. The presence of these occult neoplastic cells, as well as the presence of benign tumors, may in part contribute to cancer hormesis mechanisms acting to globally suppress malignant tumor development. 

Hormesis is a biologically conserved response, and despite the different mechanisms responsible for the biphasic response in biological systems, there is a consistent quantitative effect, which is a maximum of 30–60% response compared to controls [13]. How could these relatively modest effects modulate cancer risk? We posit that cancer hormesis manifests over long time periods; it is a modest, yet consistent, biological effect exerted over the entire history of neoplasia in a host. Such a history can span decades for many forms of cancer. Thus, cancer hormesis affects tumorigenesis to inhibit the lifetime risk of cancer and cancer progression, dependent upon the specific relationship between cancer risk and the number of neoplastic cells represented in the hormesis curve (Figure 1). Therefore, it is expected that cancer hormesis has a more potent effect on cancers that develop over long time periods in adults rather than on childhood tumors that develop within short time periods. Furthermore, given the relatively modest effects of hormesis (i.e., 30–60% response), it is expected that sporadic cancers are affected more by concomitant resistance than hereditary cancer syndromes with high penetrance. Thus, hereditary mutations with high penetrance (e.g., familial adenomatous polyposis) would be expected to exert cellular and molecular effects of a magnitude that cannot effectively be repressed by cancer hormesis. In addition, there may be a threshold below which a very small number of neoplastic cells do not confer any resistance to the development and spread of more malignant neoplastic phenotypes. 

## 3. Hypothesis

The principles of hormesis and of concomitant resistance are well-known and have been extensively discussed in the literature. In particular, the relationship between concomitant resistance and cancer has been explored and is known to have clinical relevance. The main objective of the current manuscript is not simply to repeat what is already known about these phenomena. However, there has not been a theoretical evaluation of concomitant resistance from the perspective of hormesis. Furthermore, there has not been consideration of the possibility of combining the two paradigms to understand advanced tumorigenesis as a failure of a repressive hormesis response. Thus, our objective here is to expand the understanding of concomitant resistance by interpreting it as a form of hormesis. What then is our formal hypothesis for which we propose approaches to test?

The novel hypothesis presented here is that concomitant resistance in cancer is a form of hormesis, referred to as “cancer hormesis.” We posit that the presence of benign neoplasms, or less malignant tumors (e.g., “indolent,” very slow-growing cancers with limited or no metastatic potential), can be protective against the formation of more malignant neoplasms, including metastases. If this hypothesis is correct, the presence of low-level (relatively benign) neoplasia confers greater overall resistance against cancer morbidity and mortality than a neoplasia-free condition. Therefore, in the context of a hormesis-associated response, maximal resistance to highly malignant tumors is achieved in the presence of low-level neoplasia.

If this hypothesis is correct, it suggests approaches for testing. In theory, exogenously induced cancer hormesis that utilizes engineered artificial tumors could be designed to produce an inhibitory effect, possibly exceeding the endogenous hormesis 30–60% response, and may therefore be effective even against hereditary cancer syndromes. Therefore, we further hypothesize, as a theoretical exercise for epistemological purposes, that an artificial benign primary tumor that induces natural concomitant resistance mechanisms can be a therapeutic option to inhibit malignancy and metastasis. This approach could even be used to inhibit the development of primary tumors in patients at high risk for malignant tumorigenesis, such as individuals with hereditary cancer syndromes with high penetrance. A slow-growing or, optimally, quiescent benign artificial tumor that inhibits endogenous malignant growth may therefore be a long-term therapeutic or preventive strategy. 

## 4. Testing the Hypothesis and Therapeutics

How can the hypothesis be tested? First, as described above, we expect that cancer hormesis would be observed for sporadic cancers that take decades to develop. This assumption can be tested by comparing autopsy results of cancer patients versus those of individuals without cancer. Typically, autopsy results of individuals without diagnosed cancer reveal that most adults possess “occult lesions”, neoplasms that likely would never have developed into cancer during the lifetime of the host [8]. If the cancer hormesis hypothesis is correct, individuals with cancer (or certain types of cancer) are expected to differ from individuals without diagnosed cancer, quantitatively and/or qualitatively, with respect to the occult lesions discovered upon autopsy. Thus, individuals with cancer are expected to exhibit fewer benign lesions compared to individuals without cancer (Figure 1). Further, any benign neoplasms discovered in individuals with cancer are expected to exhibit a lesser manifestation of the mechanisms [7,8,11] of concomitant resistance compared to those lesions found in individuals without cancer. Thus, individuals with cancer would quantitatively have fewer occult neoplasms and any such benign tumors that are discovered would qualitatively differ from those observed in individuals without cancer.

The same observation is expected for animal cancer models. If we consider murine experimental models that develop sporadic cancer, or cancer induced by diet or chemical carcinogens with limited penetrance, we expect mice that develop cancer to have fewer occult lesions compared to mice without cancer. Also, mice with cancer that develop metastases are expected to exhibit fewer benign neoplasms compared to those in which metastatic development is repressed. This latter finding would suggest that these benign tumors can inhibit metastatic progression in the same manner as primary cancers, although the inhibitory effect of primary cancers is expected to be stronger than that of benign neoplasms. 

In summary, a specific level of occult lesions would represent the positive end of a biphasic hormesis response, when the development of primary cancers and metastases is inhibited. Furthermore, modulation, in in vitro or in vivo studies, of molecular and cellular mechanisms (e.g., immune responses, angiogenesis, and signaling pathways contributing to cell growth) known to mediate concomitant resistance should be able to mimic/promote or inhibit concomitant resistance, dependent upon whether these mechanisms are up- or down-regulated.

To better understand the approaches for testing our hypothesis, a general schematic of endogenous and exogenous cancer hormesis is proposed (Figure 2A). A specific level of “occult” neoplastic cells can inhibit primary tumor formation (and possibly metastasis), a primary tumor can inhibit metastasis, and an exogenous artificial tumor can be used to modulate endogenous tumorigenesis (Figure 2B). If exogenous systems, utilizing (inducing) endogenous mechanisms of cancer hormesis, can inhibit cancer development, this would support the validity of endogenous cancer hormesis in the human patient.

For these animal model studies, we propose the engineering of artificial tumor cells to form benign, slow-growing, or quiescent tumors (Figure 2B). These artificial tumors would be implanted into mice in the appropriate organs, similar to orthotopic mouse models [24,25,26,27]. We note that previous work from other researchers demonstrated that injecting two tumors into mice retards the growth of the second tumor, compared to single tumor controls [9], supporting the general theoretical idea of the artificial tumor approach.

Thus, these engineered cells will possess the ability to stimulate the immune system and secrete anti-angiogenic and anti-proliferative factors. Further, if the artificial tumors are derived from, or at least directly modeled on, examples of endogenous benign tumors, they may induce concomitant resistance via additional, heretofore unknown, mechanisms that remain to be determined. The artificial tumor approach can be tested in both (a) a general design with broad efficacy against various cancer types, and (b) a design specifically tailored for the type of cancer a particular mouse model represents (e.g., colon, breast, prostate, lung, brain, etc.). Effects on overall tumorigenesis and metastasis will be measured and evaluated. In addition, the potential mechanisms by which the artificial tumor cells inhibit endogenous tumorigenesis will be determined. The data generated from each round of experiments will be used, in an iterative fashion, to redesign the artificial tumor for increased effectiveness. The approach is expected to reduce morbidity and mortality from cancer in the animal cancer models. Furthermore, the artificial tumor cells will contain a “fail-safe” suicide gene activation cassette, similar to a system established for therapeutic human pluripotent cells [28] so that the artificial tumor cells can be eliminated from the host if necessary. 

A double-edged problem to be considered in the proposed approach, if it is applied to human patients, is the host immune system response. First, artificial tumors must be designed so as not to be attacked and destroyed by the patient’s immune system. The artificial tumor cells can be engineered from the patient’s cells, or other mechanisms to evade immune destruction can be used. Animal model experiments can be used to address this question, in order to develop the most effective approach for dealing with this issue. A balance must be obtained in that the artificial tumor must be recognized by the immune system to trigger the mechanisms of concomitant resistance involving immunity [10], but, at the same time, the exogenous tumor should not be destroyed by the host immune response. Examples of endogenous tumors that possess such dual properties, and that trigger concomitant resistance hormesis, can be used to create artificial tumors with similar characteristics. Careful design concerning host immunity must be a fundamental component of the iterative process required to make this approach effective (Figure 2B). Another issue is whether the artificial tumor would be negatively affected by the same mechanisms (e.g., anti-proliferative and anti-angiogenic secreted factors) that would repress endogenous neoplasia. This same question has been asked of endogenous primary tumors; data suggest that the primary tumors produce substances, including certain amino acids, which are protective against some concomitant resistance mechanisms [3]. This finding underscores the importance of modeling the artificial system on endogenous primary neoplasms, so that the artificial tumors express the same combination of factors that repress endogenous malignant growth but are also self-protective. Further, this issue raises the question of whether the ability of secondary neoplasms to escape from concomitant resistance in the natural setting is due to mutations that allow these secondary tumors to produce the same protective factors as the primary neoplasm.

Another problem for the proposed therapeutic approach may arise in cases when cancer patients develop immunocompromised conditions. However, as mentioned above, concomitant resistance is observed in immunodeficient animal models, and this suggests that additional mechanisms play a key role in this process [7]. The possibility that the artificial tumor, designed to evade destruction by the host, may exhibit increased potential for growth in immunocompromised patients can be addressed through the aforementioned “fail-safe” design that allows for the destruction of the engineered cells [28]. All of these issues need to be evaluated empirically via animal cancer models.

Another question that needs to be experimentally addressed is the timing of the intervention. If endogenous concomitant resistance depends on modest effects exerted over long periods, what timeframe is required to achieve positive preventive effects of an artificial tumor system utilizing the hormesis approach? This question again needs to be addressed experimentally, with appropriate animal cancer models. A key point is that engineered artificial tumors, built upon an iterative design system, can manifest cancer-suppressive properties more potent than those of endogenous concomitant resistance mechanisms. The cells of the artificial tumor will be specifically engineered to maximize the proposed and identified mechanisms mediating endogenous concomitant resistance [7]. Therefore, instead of modest effects of cancer hormesis exerted over longer timeframes, the objective will be to design cells in which a stronger set of hormesis mechanisms induce anti-cancer effect within a shorter, more clinically relevant, timeframe. 

In addition to the ethical and regulatory issues involved in the artificial tumor approach (see below), another objection would be to question the necessity of this approach, and instead simply utilize for therapy the individual mechanisms that mediate concomitant resistance in cancer. Indeed, all the mechanisms reviewed above are potential therapeutic approaches in the clinical space. However, a well-designed artificial tumor approach might be far more effective than individual therapies because it simultaneously incorporates several mechanisms of action, some of which might not be yet identified. Therefore, instead of approaching the clinical use of concomitant resistance in a piecemeal fashion, applying one known mechanism at a time, the concept of cancer hormesis can be leveraged to design an artificial, benign, safe neoplastic entity that inhibits endogenous tumorigenesis through multiple mechanisms. Thus, the artificial tumor approach is most akin to naturally occurring concomitant resistance. 

Similar to the use of anti-cancer combination therapy to prevent the development of resistance, simultaneously attacking tumorigenesis by several mechanisms induced by an engineered tumor may have a better long-lasting outcome. Whereas it is true that combination therapy can also be performed by treating the patient with several individual modalities mimicking concomitant resistance mechanisms, the application of the artificial tumor approach would be more efficient. Furthermore, by modeling the artificial tumor after benign non-symptomatic endogenous neoplasms, the artificial tumor approach may cause fewer systemic side effects compared to a cocktail of therapeutic agents that inhibit or stimulate physiological processes. In addition, combination therapy approaches are hindered by many obstacles, including (a) the necessity to determine the dose for each component of the therapy alone and in combination with other components, (b) the complexity of clinical trial design, and (c) the fact that various modalities are offered by different pharmaceutical companies and, therefore, their combined application requires legal and commercial agreements. 

Considering the molecular and cellular mechanisms, including signaling pathways, which are involved in the concomitant resistance response [5,6,7,8,9,10] it is possible to combine the engineered tumor approach synergistically with other more traditional therapies. Angiogenesis inhibitors or immune checkpoint inhibitors can be utilized to enhance the engineered tumor’s ability to suppress endogenous tumor growth via the suppression of angiogenesis or promotion of anti-tumor immune responses. 

In addition, other molecular pathways of concomitant resistance involve signaling pathways involved in the control of cell proliferation. For example, tyrosine isomers can mediate concomitant resistance through MAP/ERK inhibition and STAT3 inactivation, as well as via effects on cell cycle checkpoints [7]. Thus, the ERK inhibitor ulixertinib or the STAT3 inhibitors BB1608 and Celecoxib could be useful in combination with the artificial tumor approach. Furthermore, therapeutic agents that affect cell cycle checkpoints [29] may also prove useful in a combination approach along with artificial tumor strategy. Experiments with animal cancer models can be used to evaluate such combination approaches and further explore the molecular mechanisms of concomitant resistance to identify additional targets and pathways for synergistic therapies. 

The retrospective human and mouse work (first part of our proposed hypothesis testing) is methodologically doable now; the more difficult part of the proposed studies would the prospective mouse work using the artificial tumor system. We believe that this mouse model work is, theoretically, methodologically possible, albeit difficult. In addition, despite possible methodological difficulties in performing some of the proposed experiments we believe that there is epistemological benefit to present the hypothesis, and the theoretical approaches for testing, to help illustrate the cancer hormesis concept, stimulate discussion and debate, and prompt other researchers to consider attempting such studies. It is possible other researchers will have, or develop, the capability of performing these studies; alternatively, they may devise other approaches that are methodologically easier while obtaining the required data. Therefore, we believe that there is utility in introducing the experimental schematics to the scientific community for consideration and evaluation even if the methodology presents certain challenges.

## 5. Ethics for Potential Clinical Use

There will be ethical and regulatory issues in establishing the engineered tumor approach in patients. This issue can be approached in several ways. First, this manuscript, including the sections on proposed clinical use, is presented as a theoretical model, i.e., as a tool of understanding. As noted above, the artificial tumor approach can be analyzed, and its implications considered from an epistemological perspective, to learn more about cancer hormesis and concomitant resistance. Second, we must consider that radical measures and therapies can be proposed for terminally ill patients for whom all other therapeutic options have failed. The principles of “extended access” and “compassionate use” for novel therapies, including those not thoroughly researched and proven, in terminally ill patients are well established. As part of the personal autonomy of the patient, informed consent for experimental approaches can be considered ethically appropriate under specific circumstances. Third, the incorporation of “fail-safe” and “self-destruct” mechanisms into the engineered cells [28] will allow for the elimination of the engineered cells, if necessary, in closely monitored patients. At any point in which the artificial tumor approach is deemed a potential threat to patient safety, the cells can be destroyed by relevant mechanisms.

## 6. Discussion and Conclusions

This paper proposes that a type of hormesis involving neoplasia—cancer hormesis—affects the lifecycle of tumorigenesis. Low-level occult neoplasia may inhibit the development of more advanced malignancy, by stimulating or repressing the host and tumor functions, as part of the positive end of a biphasic hormesis response to neoplastic stress. Similarly, primary tumors can repress secondary malignancies, which also fits with the cancer hormesis paradigm.

Aspects of resource competition, game theory, and horizontal versus vertical transmission play a role in the adaptive responses of neoplastic cells to hormesis and concomitant resistance [30]. Considering the existence of mutations in neoplastic cells, the primary tumor is not genetically identical to normal host cells, and even more important for cancer hormesis, secondary lesions are typically genetically different from primary lesions. Therefore, competition between classes of neoplastic cells can therefore be adaptive in the neo-Darwinian sense. Tumorigenesis involves the release of normal cellular cooperative controls in favor of “selfish” uncontrolled growth and genetic expansion. In this sense, evolutionary game theory and differential transmission play a role in cancer hormesis; metastases favor horizontal transmission of the neoplasia throughout the host, whereas concomitant resistance favors the vertical transmission of the primary tumor in its original location over time while suppressing horizontal transmission. These differences in adaptive interests contribute to the evolutionary competition that drives cancer hormesis, and thus can inform the iterative design of clinically relevant concomitant resistance approaches for cancer prevention and treatment.

Finally, as a theoretical exercise, to illustrate the cancer hormesis paradigm, a therapeutic system utilizing an engineered artificial tumor, designed to mimic the positive hormesis effects of “occult” neoplastic lesions (or primary tumors), is proposed. This approach can be utilized to prevent the development of cancer in high-risk individuals and/or to inhibit or repress metastases in individuals with cancer. In the latter case, this approach can allow for the removal of the primary tumor without a subsequent increased risk of metastases due to the elimination of the inhibitory activity of the primary tumor. Considerations of cancer hormesis and the general phenomenon of concomitant resistance can have even more immediate medical consequences with respect to decisions of whether to resect the primary tumor in different clinical scenarios [4].

In summary, a better understanding of cancer hormesis can lead to novel approaches that reduce morbidity and mortality from human cancer. 

## Figures and Tables

**Figure 1 life-14-00401-f001:**
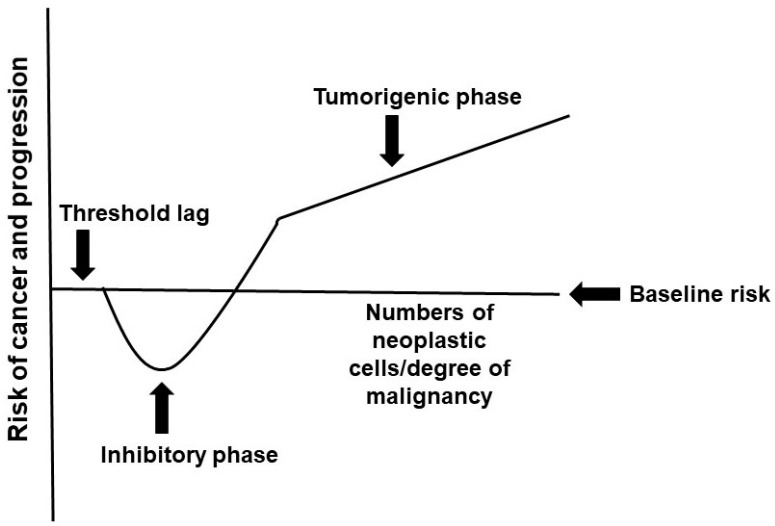
Cancer Hormesis. An individual with no neoplastic cells or neoplasia under the “lag threshold” will exhibit a baseline risk of cancer and progression (y-axis = 0). A neoplastic burden beyond the threshold will stimulate a positive hormesis effect of decreasing cancer risk and/or risk of further progression. However, beyond a certain point, advanced neoplasia will markedly increase the risk of tumorigenesis, including metastases.

**Figure 2 life-14-00401-f002:**
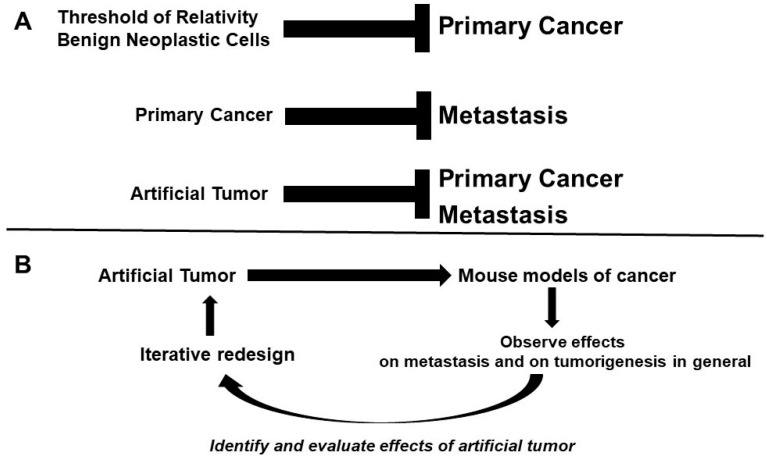
Therapeutic Approach. (**A**) The general schematic of cancer hormesis upon which therapeutics can be designed. (**B**) Iterative scheme for testing the artificial tumor approach in animal cancer models. Cells will be engineered to grow as benign, slow-growing tumors with limited maximum size potential, coupled with the ability to stimulate the immune system, and secrete anti-angiogenic factors, and anti-proliferative factors, etc. The artificial tumor approach can be tested in a design with broad efficacy against diverse cancer types or can be tailored for specific types of cancer in relevant murine cancer models. Effects on overall tumorigenesis and metastasis will be measured. Potential mechanisms of action that affect tumor growth and dissemination will also be ascertained. Data from each round of experiments will be used, in an iterative fashion, to redesign the artificial tumor for increased effectiveness. Ultimately, reduced morbidity and mortality is expected.

## Data Availability

No new data were created.

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
