# Peer review of "Hypothesis: Cancer Hormesis and Its Potential for Cancer Therapeutics"

_life, 2024, doi:10.3390/life14030401_

Round 1

Reviewer 1 Report

Comments and Suggestions for Authors

In this manuscript, the author proposes a type of cancer hormesis, that can be leveraged for therapeutic purposes, and corresponding strategies for testing the hypothesis are also explained. It is a bold and innovative idea to use primary tumors with safety control for high-risk patients' treatment, but promising. Although the authors have described the tested approaches for this hypothesis and therapeutics, the major point that I am concerned about is do you have some preliminary results to support your hypothesis in the mice model. If yes, you’d better show it in the manuscript. Another small stuff is that 1) the case of the first letter of titles should be kept consistent, and 2) it is unnormal to cite one piece of literature through a total paragraph ([12] in Discussion and Conclusion). 

Comments on the Quality of English Language

 Moderate editing of English language required

Author Response

We unfortunately have no preliminary data for any of the proposed experimental work of this hypothesis paper, including for the proposed mouse model studies. We note that the new reference [9] refers to data in the literature in which injecting two tumors into mice retards the growth of the second tumor, compared to controls, which supports the general theoretical idea of the artificial tumor approach. A sentence on this has been added to the testing the hypothesis section (lines 241-243). However, despite having no preliminary data of our own, we believe that the mouse model work is methodologically possible, albeit difficult.  It would require, as the text indicates, engineering cells (e.g., stable transection, CRISPR, etc.) to possess the required attributes for concomitant resistance and implanting these into the appropriate mouse strains in the tissue of interest, akin to orthotopic mouse experiments. While we do not have the capability of performing these studies, we hope this paper will stimulate interest for others to pursue this line of inquiry. 

With respect to the minor issues, we corrected letter case for the reference titles for consistency, and reference [12] – now reference [30] - is cited once in the relevant paragraph.

Reviewer 2 Report

Comments and Suggestions for Authors

The manuscript „Hypothesis: Cancer Hormesis and its Potential for Cancer Therapeutics” by Michael Bordonaro  and Darina Lazarova presents an interesting hypothesis:  that the mechanisms of concomitant resistance in cancer can be explained or associated with the hormesis effect. Indeed, the resistence to secondary tumor or  metastasis in many types of cancer have been observed and  immunological and non-immunological mechanisms have been proposed based on experimental data. The authors also suggest that occult lesions reported after post-mortem examinations of healthy individuals  could represent the evidence of a beneficial hormesis response in carcinogenesis by stopping the tumor development. They propose as future therapeutic  approach the implantation of  engineered artificial tumor,  with slow-growing rate, or quiescent tumors in animal models (to test the mechanism) or patients with certain type of cancer. The ethical issues are also discussed.

The hypothesis is  relatively well argued,   however the authors should have cited more relevant scientific works that led them to this hypothesis. In my opinion the hypothesis of cancer hormesis would be an interesting and exciting  point of view.

I ask the authors to add more citations in the arguments they made.

Author Response

We have added 18 additional citations, ranging in year from 1982-2023.  These include papers on the basic theories of concomitant resistance and hormesis, the possible role of hormesis in cancer therapy, as well as references to orthotopic mouse models methodologically akin to some of the work proposed in our paper (i.e., implanting artificial tumors). These new references are marked in red font in the References section of the revised text and are reference numbers 1,3,4,9,11,14-22, and 24-27.

These added references have broadened the depth of our discussion of these issues and provide the reader with citations to allow for more detailed investigation of these topics.

Reviewer 3 Report

Comments and Suggestions for Authors

Comments to life-2894318

Thank you for sending me this manuscript by Michael Bordonaro and Darina Lazarova to review. Overall, the authors explain some ideas about less tumorigenic neoplastic cells helping to inhibit secondary lesion, metastases or other affections. Authors hypothesize that concomitant resistance is a form of hormesis, a biphasic dose response in which a stimulus has a positive and/or stimulatory effect at low dosages and a negative, inhibitory, and/or toxic effect at higher dosages.

There are issues that are listed in order, as follow:

1) please add more information to the introduction.

2) writing style should be improved, it is a bit difficult to follow the idea since there are very repetitive words within a paragraph (for example: occult neoplastic cells in page 3, paragraph 1).

3) the central hypothesis is described across the different sections (Cancer hormesis section). I do not know what part of the sections the cancer hormesis is included in.

4) font size in figure 2 legend does not match to the figure 1. Please homogenize the legend´s font size and style.

5) some parts of the hypothesis are a bit complicated to be proved or to start investigations on this topic.

Comments on the Quality of English Language

text editing is required

Author Response

There are issues that are listed in order, as follow:

1) please add more information to the introduction.

Additional material was added to the Introduction (red font). The Introduction has thus been lengthened by ~ 85%, resulting in a mor in depth discussion of concomitant resistance and its mechanisms.

2) writing style should be improved, it is a bit difficult to follow the idea since there are very repetitive words within a paragraph (for example: occult neoplastic cells in page 3, paragraph 1).

We have edited the work for increased clarity, including for the cited example.

3) the central hypothesis is described across the different sections (Cancer hormesis section). I do not know what part of the sections the cancer hormesis is included in.

The manuscript is structured so that the Introduction discusses the basic principles of concomitant resistance and the proposed mechanisms for this resistance. This is followed by the section Cancer Hormesis that discusses the basic principles of hormesis followed by a broad theoretical exploration of our idea of cancer hormesis, and how it fits with the general hormesis mechanism.  The Hypothesis section formally states the hypothesis, building on the previous two sections. We then proceed to discuss testing the hypothesis and implications.

            Thus, the cancer hormesis idea is first introduced in the Cancer Hormesis section and the formal statement of the hypothesis is in the Hypothesis section. We have moved around some text (e.g., introducing the artificial tumor approach) between the Cancer Hormesis and Hypothesis sections to better distinguish those sections. We have also eliminated the word “hypothesis” from the Cancer Hormesis section to stress that this section has as its objective introducing the idea of cancer hormesis and how its fits with the general hormesis paradigm, while the formal hypothesis itself is stated in the Hypothesis section. In this manner, the manuscript flows in the logical sequence of general introduction and concomitant resistance, hormesis and how the introduced concept of cancer hormesis fits into the hormesis mechanism, formal statement of the hypothesis and mention of the artificial tumor approach, then testing the hypothesis, followed implications and conclusions.

Thus, to summarize, the general idea of cancer hormesis is introduced in the Cancer Hormesis section, while the formal, definitive statement of the hypothesis is presented in the Hypothesis section, and it is this formal hypothesis which forms the basis of our proposed experiments.

4) font size in figure 2 legend does not match to the figure 1. Please homogenize the legend´s font size and style.

This was corrected.

5) some parts of the hypothesis are a bit complicated to be proved or to start investigations on this topic.

While we agree that some parts of the hypothesis testing may be difficult, we still believe it is useful to discuss the possibilities as part of the theoretical considerations of this manuscript.            The retrospective human and mouse work (first part of hypothesis testing) is for the most part methodologically doable now; the more difficult part of the proposed studies would the prospective mouse work with the artificial tumor system. We believe this mouse model work is, theoretically, methodologically possible, albeit difficult.  It would require, as the text indicates, engineering cells (e.g., stable transection, CRISPR, etc.) to possess the required attributes and implanting into the appropriate mouse strains in the tissue of interest, akin to orthotopic mouse experiments.  

In addition, despite possible methodological difficulties in performing some of the proposed experiments we believe that there is epistemological benefit to present the hypothesis and theoretical approaches for testing to help illustrate the cancer hormesis concept, stimulate discussion and debate, and prompt other researchers to consider attempting such studies. Thus, in the paper we introduce the artificial tumor approach as “as a theoretical exercise for epistemological purposes.”

While we do not have the capability of performing these studies, we hope this paper will stimulate interest for others to pursue this line of inquiry. It is possible other researchers will have or develop the capability of performing these studies; alternatively, they may devise other approaches that are methodologically easier while obtaining the required data.  Therefore, we believe that there is utility in introducing the experimental schematics to the scientific community for consideration and evaluation even if the methodology presents certain challenges.

A relevant paragraph about this issue has been added to the end of the section on testing the hypothesis (lines 363-375).

Reviewer 4 Report

Comments and Suggestions for Authors

This paper introduced the concept of cancer Hormesis and concomitant resistance, indicating a new theory of cancer treatment. 

I see the article type is labelled as "hypothesis". But still, I feel this paper did not meet the requirement to be considered as a research article. It does not have experimental data, nor analysis of clinical data. All is based on deduction but no verification. I doubt if it could make any solid contribution to this research area at all.

Author Response

It is difficult to answer this critique other than to state:

  1. The other three reviewers saw considerable benefit of this paper as contributing to the field. Therefore, there is no consensus with respect to the conclusion of this reviewer.
  2. The revised paper discusses in more depth the clinical and experimental data in the literature, as part of the expanded Introduction. Therefore the stated lack of analysis has been corrected.
  3. Our understanding of a hypothesis paper is that it presents a novel hypothesis derived from findings from the literature combined with insights and speculation from the authors.  This is what this paper does. It is not clear from our experience that it is mandatory for a hypothesis paper to contain experimental data from the paper’s authors. Verification is what occurs after the publication of a hypothesis paper, not before; if the idea were verified, it would no longer be a hypothesis, but a straight research manuscript.

Round 2

Reviewer 1 Report

Comments and Suggestions for Authors

N/A

Comments on the Quality of English Language

N/AN/A

Reviewer 2 Report

Comments and Suggestions for Authors

thank you  to the authors for their response  and for the changes made to the manuscript. The manuscript can be published in this form.

Reviewer 3 Report

Comments and Suggestions for Authors

suggestions were addressed and issues were fixed.

Reviewer 4 Report

Comments and Suggestions for Authors

No further comments.